# Large Batch Size Training of Neural Networks with Adversarial Training and Second-Order Information

## Abstract

Stochastic Gradient Descent (SGD) methods using randomly selected batches are widely-used to train neural network (NN) models. Performing design exploration to find the best NN for a particular task often requires extensive training with different models on a large dataset, which is very computationally expensive. The most straightforward method to accelerate this computation is to distribute the batch of SGD over multiple processors. To keep the distributed processors fully utilized requires commensurately growing the batch size; however, large batch training often times leads to degradation in accuracy, poor generalization, and even poor robustness to adversarial attacks. Existing solutions for large batch training either significantly degrade accuracy or require massive hyper-parameter tuning. To address this issue, we propose a novel large batch training method which combines recent results in adversarial training (to regularize against "sharp minima") and second order optimization (to use curvature information to change batch size adaptively during training). We extensively evaluate our method on Cifar-10/100, SVHN, TinyImageNet, and ImageNet datasets, using multiple NNs, including residual networks as well as smaller networks for mobile applications such as SqueezeNext. Our new approach exceeds the performance of the existing solutions in terms of both accuracy and the number of SGD iterations (up to 1% and 5×, respectively). We emphasize that this is achieved without any additional hyper-parameter tuning to tailor our proposed method in any of these experiments.

## 1 Introduction

Finding the right NN architecture for a particular application requires extensive hyper-parameter tuning and architecture search, often times on a very large dataset. The delays associated with training NNs is often the main bottleneck in the design process. One of the ways to address this issue to use large distributed processor clusters; however, to efficiently utilize each processor, the portion of the batch associated with each processor (sometimes called the mini-batch) must grow correspondingly. In the ideal case, the hope is to decrease the computational time proportional to the increase in batch size, without any drop in generalization quality. However, large batch training has a number of well known draw backs. These include degradation of accuracy, poor generalization, and poor robustness to adversarial perturbations (Keskar et al., 2016; Yao et al., 2018).

In order to address these drawbacks, many solutions have been proposed (Goyal et al., 2017; You et al., 2017; Devarakonda et al., 2017; Smith et al., 2017; Jia et al., 2018). However, these methods either work only for particular models on a particular dataset, or they require massive hyper-parameter tuning, which is often times not discussed in the presentation of results. Note that while extensive hyper-parameter turning may result in good result tables, it is antithetical to the original motivation of using large batch sizes to reduce training time.

One solution to reduce the brittleness of SGD to hyper-parameter tuning is to use second-order methods. Full Newton method with line search is parameter-free, and it does not require a learning rate. This is achieved by using a second-order Taylor series approximation to the loss function, instead of a first-order one as in SGD, to obtain curvature information. Schaul et al. (2013); Xu et al. (2017a;b) show that Newton/quasi-Newton methods outperform SGD for training NNs. However, their re-

sults only consider simple fully connected NNs and auto-encoders. A problem with second-order methods is that they can exacerbate the large batch problem, as by construction they have a higher tendency to get attracted to local minima as compared to SGD. For these reasons, early attempts at using second-order methods for training convolutional NNs have so far not been successful.

Ideally, if we could find a regularization scheme to avoid local/bad minima during training, this could resolve many of these issues. In the seminal works of El Ghaoui & Lebret (1997); Xu et al. (2009), a very interesting connection was made between robust optimization and regularization. It was shown that the solution to a robust optimization problem for least squares is the same as the solution of a Tikhonov regularized problem (El Ghaoui & Lebret, 1997). This was also extended to the Lasso problem in Xu et al. (2009). Adversarial learning/training methods, which are a special case of robust optimization methods, are usually described as a min-max optimization procedure to make the model more robust. Recent studies with NNs have empirically found that robust optimization usually converges to points in the optimization landscape that are flatter and are more robust to adversarial perturbation (Yao et al., 2018).

Inspired by these results, we explore whether second order information regularized by robust optimization can be used to do large batch size training of NNs. We show that both classes of methods have properties that can be exploited in the context of large batch training to help reduce the brittleness of SGD with large batch size training, thereby leading to significantly improved results.

## MAIN CONTRIBUTIONS

In more detail, we propose an adaptive batch size method based on curvature information extracted from the Hessian, combined with a robust optimization method. The latter helps regularize against sharp minima, especially during early stages of training. We show that this combination leads to superior testing performance, as compared to the proposed methods for large batch size training. Furthermore, in addition to achieving better testing performance, we show that the total number of SGD updates of our method is significantly lower than state-of-the-art methods for large batch size training. We achieve these results without any additional hyper-parameter tuning of our algorithm (which would, of course, have helped us to tailor our solution to these experiments). Here is a more detailed itemization of the main contributions of this work:

- We propose an Adaptive Batch Size method for SGD training that is based on second order information, computed by backpropagating the Hessian operator. Our method automatically changes the batch size and learning rate based on Hessian information. We state and prove a result that this method is convergent for a convex problem. More importantly, we empirically test the algorithm for important non-convex problems in deep learning and show that it achieves equal or better test performance, as compared to small batch SGD (We refer to this method as ABS).
- We propose a regularization method using robust training by solving a min-max optimization problem. We combine the second order adaptive batch size method with recent results of Yao et al. (2018), which show that robust training can be used to regularize against sharp minima. We show that this combination of Hessian-based adaptive batch size and robust optimization achieves significantly better test performance with little computational overhead (we refer to this Adaptive Batch Size Adversarial method as ABSA).
- We test the proposed strategies extensively on a wide range of datasets (Cifar-10/100, SVHN, TinyImageNet, and ImageNet), using different NNs, including residual networks. Importantly, we use the *same hyper-parameters* for all of the experiments, and we do not perform any kind of tuning of our hyper-parameters to tailor our results. The empirical results show the clear benefit of our proposed method, as compared to the state-of-the-art. The proposed algorithm achieves equal or better test accuracy (up to 1%) and requires significantly fewer SGD updates (up to $5\times$).
- We empirically show that we can use a block approximation of the Hessian operator (i.e. the Hessian of the last fewer layers) to reduce the computational overhead of backpropagating the second order information. This approximation is especially effective for deep NNs.

While a number of recent works have discussed adaptive batch size or increasing batch size during training (Devarakonda et al., 2017; Smith et al., 2017; Friedlander & Schmidt, 2012; Balles et al., 2016), to the best of our knowledge this is the first paper to introduce Hessian information and adversarial training in adaptive batch size training, with extensive testing on many datasets.

LIMITATIONS

We believe that it is important for every work to state its limitations (in general, but in particular in this area). We were particularly careful to perform extensive experiments and repeated all the reported tests multiple times. We test the algorithm on models ranging from a few layers to hundreds of layers, including residual networks as well as smaller networks such as SqueezeNext.

An important limitation is that second order methods have additional overhead for backpropagating the Hessian. Currently, most of the existing frameworks do not support (memory) efficient backprop-agation of the Hessian (thus providing a structural bias against these powerful methods). However, the complexity of each Hessian matvec is the same as a gradient computation (Martens, 2010). Our method requires Hessian spectrum, which typically needs ten Hessian matvecs (for power method iterations to reach a tolerance of 1e-2). Thus, the benefits that we show in terms of testing accuracy and reduced number of updates do come at a cost (see Table 3 for details). We measure this addi-tional overhead and report it in terms of wall clock time. Furthermore, we (empirically) show that this power iteration needs to be done only at the end of every epoch, thus significantly reducing the additional overhead.

Another limitation is that our theory only holds for convex problems (under certain smoothness as-sumptions). Proving convergence for non-convex setting requires more involved analysis. Recently, Ward et al. (2018) has provided interesting theoretical guarantees for AdaGrad (Duchi et al., 2011) in the non-convex setting. Exploring a similar direction for our method is of interest for future work. Another point is that adaptive batch size, prevents one from utilizing all of the processes, as compared to using large batch throughout the training. However, a large data center can handle and accommodate a growing number of requests for processor resources, which could alleviate this.

## 2 RELATED WORK

Optimization methods based on SGD are currently the most effective techniques for training NNs, and this is commonly attributed to SGD's ability to escape saddle-points and "bad" local min-ima (Dauphin et al., 2014).

The sequential nature of weight updates in synchronous SGD limits possibilities for parallel com-puting. In recent years, there has been considerable effort on breaking this sequential nature, through asynchronous methods (Zhang et al., 2015) or symbolic execution techniques (Maleki et al., 2017). A main problem with asynchronous methods is reproducibility, which, in this case, depends on the number of processes used (Zheng et al., 2016; Agarwal & Duchi, 2011). Due to this issue, recently there have been attempts to increase parallelization opportunities in synchronous SGD by using large batch size training. With large batches, it is possible to distribute more efficiently the computations to parallel compute nodes (Gholami et al., 2018a), thus reducing the total training time. However, large batch training often leads to sub-optimal test performance (Keskar et al., 2016; Yao et al., 2018). This has been attributed to the observation that large batch size training tends to get attracted to local minima or sharp curvature directions, which are not robust to (possible) mismatch between training and testing curves (Keskar et al., 2016). A full understanding of this, however, remains elusive.

There have been several solutions proposed for alleviating the problem with large batch size training. The first notable work here is Goyal et al. (2017), where it was shown that by scaling the learning rate, it is possible to achieve the same testing accuracy for large batches. In particular, ResNet-50 model was tested on ImageNet dataset, and it was shown that the baseline accuracy could be recovered up to a batch size of 8192. However, this approach does not generalize to other networks such as AlexNet (You et al., 2017), or other tasks such as NLP. In You et al. (2017), an adaptive learning rate method (called LARS) was proposed which allowed scaling training to a much larger batch size of 32K with more hyper-parameter tuning. Another notable work is Smith et al. (2017) (and also Devarakonda et al. (2017)), which proposed a hybrid increase of batch size and learning rate to accelerate training. In this approach, one would select a strategy to "anneal" the batch size during the training. This is based on the idea that large batches contain less "noise," and that could be used much the same way as reducing learning rate during training. More recent work Jia et al. (2018); Puri et al. (2018) proposed mix-precision method to further explore the limit of large batch training.

A recent study has shown that anisotropic noise injection could also help in escaping sharp minima (Zhu et al., 2018). The authors showed that the noise from SGD could be viewed as anisotropic, with the Hessian as its covariance matrix. Injecting random noise using the Hessian as covariance was proposed as a method to avoid sharp minima.

Another recent work by Yao et al. (2018) has shown that adversarial training (or robust optimization) could be used to "regularize" against these sharp minima, with preliminary results showing superior testing performance as compared to other methods. The link between robust optimization and regularization is a very interesting observation that has been theoretically proved in the case of Ridge regression (El Ghaoui & Lebret, 1997), and Lasso (Bertsimas et al., 2011). Shaham et al. (2015); Shrivastava et al. (2017) used adversarial training and showed that the model training using robust optimization is often times more robust to perturbations, as compared to normal SGD training. Similar observations have been made by others (Szegedy et al., 2013; Goodfellow et al., 2014).

## 3   OUR MAIN METHOD

We consider a supervised learning framework where the goal is to minimize a loss function $L(\theta)$:

$$L(\theta) = \frac{1}{N} \sum_{i=1}^{N} l(z_i, \theta), \tag{1}$$

where $\theta$ are the model weight parameters, $Z = X \times Y$ is the training dataset, and $l(z, \theta)$ is the loss for a datum $z \in Z$. Here, $X$ is the input, $Y$ is the corresponding label, and $N = |Z|$ is the cardinality of the training set. SGD is typically used to optimize Eqn. (1) by taking steps of the form:

$$\theta_{t+1} = \theta_t - \eta_t \frac{1}{|B|} \sum_{z \in B} \nabla_\theta l(z, \theta_t), \tag{2}$$

where $B$ is a mini-batch of examples drawn randomly from $Z$, and $\eta_t$ is the step size (learning rate) at iteration $t$. In the case of large batch size training, the batch size is increased to large values.

Smith & Le (2018) views the learning rate and batch size as noise injected during optimization. Both a large learning rate as well as a small batch size can be considered to be equivalent to high noise injection. This is explained by modeling the behavior of NNs as a stochastic differential equation (SDE) of the following form:

$$\frac{d\theta}{dt} = \frac{dL}{d\theta} + \epsilon(t), \tag{3}$$

where $\epsilon(t)$ is the noise injected by SGD (see Smith & Le (2018) for details). The authors then argue that the noise magnitude is proportional to $g = \eta_t(\frac{|Z|}{|B|} - 1)$. For mini-batch $|B| \ll |Z|$, the noise magnitude can be estimated as $g \approx \eta_t \frac{|Z|}{|B|}$. Hence, in order to achieve the benefits from small batch size training, i.e., the noise generated by small batch training, the learning rate $\eta_t$ should increase proportionally to the batch size, and vice versa. That is, the same annealing behavior could be achieved by increasing the batch size, which is the method used by Smith et al. (2017).

The need for annealing can be understood by considering a convex problem. When we get closer to a local minimum, a more accurate descent direction with less noise is preferable to a more noisy direction, since less noise helps converge to rather than oscillate around the local minimum. This explains the manual batch size and learning rate changes proposed in (Smith et al., 2017; Devarakonda et al., 2017). Ideally, we would like to have an automatic method that could provide us with such information and regularize against local minima with poor generalization. As we show next, this is possible through the use of second order information combined with robust optimization.

### 3.1   ADAPTIVE BATCH SIZE (ABS) BASED ON HESSIAN INFORMATION

In this section, we propose a method for utilizing second order information to adaptively change the batch size. We refer to this as the Adaptive Batch Size (ABS) method; see Alg. 1. Intuitively, using a larger batch size in regions where the loss has a "flatter" landscape, and using a smaller batch size in regions with a "sharper" loss landscape, could help to avoid attraction to local minima with poor generalization. This information can be obtained through the lens of the Hessian operator.

---

**Algorithm 1** Adaptive Batch Size (ABS) and Adaptive Batch Size Adversarial (ABSA)

---

1: **Input:**
  - Learning rate $lr$, learning rate decay steps $A$, learning rate decay ratio $\rho$
  - Initial Batch Size $B$, minimum batch size $B_{min}$, maximum batch size $B_{max}$, input $x$, label $y$.
  - Eigenvalue decreasing ratio $\alpha$, eigenvalue computation frequency $n$, i.e., after training $n$ samples compute eigenvalue, batch increasing ratio $\beta$, duration factor $\kappa$, i.e., if we compute $\kappa$ times Hessian but eigenvalue does not decrease, we would increase the batch size
  - If adversarial training is used, perturbation magnitude $\epsilon_{adv}$, perturbation ratio $\gamma$ ($\gamma_{max}$) of training data, decay ratio $\omega$, vanishing step $\tau$
2: **Initialization: Eig = None, Visiting Sample = 0**
3: **for** $t = 0, 1, \ldots$ **do**
4:     **if** $\gamma > 0$ **then**
5:         $x[: B\gamma] = adversarial(x[: B\gamma])$
6:     One step SGD updates
7:     Visiting Sample $+ =$ B
8:     **if** Visiting Sample % n is 0 **then**
9:         Current Eigen = Computing Eigen
10:         **if** Current Eigen $< \frac{1}{2}*$Eig **then**
11:             $B = \min\{B\beta, B_{max}\}$, change $lr$
12:             $\gamma = \gamma/2$, Eig = Current Eigen
13:             Duration Time = 0
14:         **else**
15:             Duration Time $+ =$ 1
16:     **if** Duration Time = $\kappa$ **then**
17:         $B = \min\{B\beta, B_{max}\}$, change $lr$
18:         Duration Time = 0
19:     **if** t in A **then**
20:         decay the learning rate by $\rho$
21:     **if** $t == \tau$ **then**
22:         $\gamma = 0$

---

We adaptively increase the batch size as the Hessian eigenvalue decreases or stays stable for several epochs (fixed to be ten in all of the experiments).

The second component of our framework is robust optimization. In the seminal works of (El Ghaoui & Lebret, 1997; Xu et al., 2009), a connection between robust optimization and regularization was proved in the context of ridge and lasso regression. In Yao et al. (2018), the authors empirically showed that adversarial training leads to more robust models with respect to adversarial perturbation. An interesting corollary was that, after adversarial training, the model converges to regions that are considerably flatter, as compared to the baseline.

Thus, we can combine our ABS algorithm with adversarial training as a form of regularization against "sharp" minima. We refer to this as the Adaptive Batch Size Adversarial (ABSA) method; see Alg. 1. In practice, ABSA is often more stable than ABS. This corresponds to solving a min-max problem instead of a normal minimization problem (Keskar et al., 2016; Yao et al., 2018). Solving this min-max problem for NNs is an intractable problem, and thus we approximately solve the maximization problem through the Fast Gradient Sign Method (FGSM) proposed by Goodfellow et al. (2014). This basically corresponds to generating adversarial inputs using one gradient ascent step (i.e., the perturbation is computed by $\Delta x = \epsilon \nabla_x l(z, \theta)$). Other possible choices are proposed by (Thakur et al., 2005; Carlini & Wagner, 2017; Moosavi-Dezfooli et al., 2016).[1]

Figure 1 illustrates our ABS schedule as compared to a normal training strategy and the increasing batch size strategy of Smith et al. (2017); Devarakonda et al. (2017). Note that our learning rate adaptively changes based on the Hessian eigenvalue in order to keep the same noise level as in the baseline SGD training. As we show in section 4, our combined approach (second order and robust optimization) not only achieves better accuracy, but it also requires significantly fewer SGD updates, as compared to Smith et al. (2017); Devarakonda et al. (2017).

## 3.2 CONVERGENCE RATE OF ABS

Before discussing the empirical results, an important question is whether using ABS is a convergent algorithm for even a convex problem. Here, we show that our ABS algorithm does converge for strongly convex problems. Based on an assumption about the loss (Assumption 2 in Appendix A), it is not hard to prove the following theorem.

---

[1]In Yao et al. (2018), similar behavior was observed with other methods for solving the robust optimization problem.

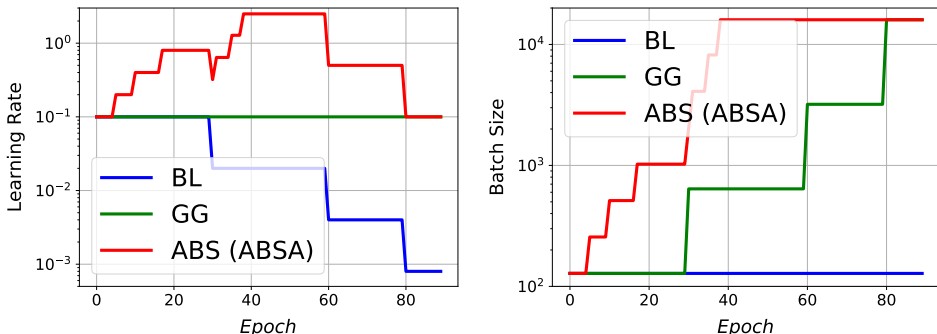

**Figure 1:** *Illustration of learning rate (left) and batch size (right) schedules of adaptive batch size as a function of training epochs based on C2 model on Cifar-10.*

**Theorem 1.** *Under Assumption 2, let assume at step $t$, the batch size used for parameter update is $b_t$, the step size is $b_t\eta_0$, where $\eta_0$ is fixed and satisfies,*

$$0 < \eta_0 \leq \frac{1}{L_g(M_v + B_{\max})}, \tag{4}$$

*where $B_{\max}$ is the maximum batch size during training. Then, with $\theta_0$ as the initilization, the expected optimality gap satisfies the following inequality,*

$$\mathbb{E}[L(\theta_{t+1})] - L_* \leq \prod_{k=1}^{t}(1 - b_k\eta_0 c_s)(L(\theta_0) - L_* - \frac{\eta_0 L_g M}{2c_s}) + \frac{\eta_0 L_g M}{2c_s}. \tag{5}$$

From Theorem 1, if $b_t \equiv 1$, the convergence rate for $t$ steps, based on equation 5, is $(1 - \eta_0 c_s)$. However, the convergence rate of Alg. 1 becomes $\prod_{k=1}^{t}(1 - b_k\eta_0 c_s)$, where $1 \leq b_k \leq B_{max}$. With an adaptive $b_t$, Alg. 1 can converge faster than basic SGD. We show empirical results for a logistic regression problem in the Appendix A, which is a simple convex problem.

## 4 Our Main Results

We evaluate the performance of our ABS and ABSA methods on different datasets (ranging from O(1E4) to O(1E7) training examples) and multiple NN models. We compare the baseline performance (i.e., small batch size), along with other state-of-the-art methods proposed for large batch training (Smith et al., 2017; Goyal et al., 2017). The two main metrics for comparison are (1) the final accuracy and (2) the total number of updates. Preferably we would want a higher testing accuracy along with fewer SGD updates. We emphasize that, for all of the datasets and models we tested, we do *not* change any of the hyper-parameters in our algorithm. We use the exact same parameters used in the baseline model, and we do not tailor any parameters to suit our algorithm. A detailed explanation of the different NN models, and the datasets is given in Appendix B.

Section 4.1 shows the result of ABS (ABSA) compared to BaseLine (BL), FB (Goyal et al., 2017) and GG (Smith et al., 2017). Section 4.2 presents the results on more challenging datasets of Tiny-ImageNet and **ImageNet**. The superior performance of our method does come at the cost of back-propagating the Hessian. Thus, in section 4.3, we discuss how approximate Hessian informatino could be used to alleviate teh costs.

### 4.1 ABS and ABSA for SVHN and Cifar

We first start by discussing the results of ABS and ABSA on SVHN and Cifar-10/100 datasets. Notice that GG and our ABS (ABSA) have different batch sizes during training. Hence the batch size reported in our results represents the maximum batch size during training. To allow for a direct comparison we also report the number of weight updates in our results (lower is better). It should be mentioned that the number of SGD updates is not necessarily the same as the wall-clock time. Therefore, we also report a simulated training time of I3 model in Appendix C.

Tables 1 and 4-7 (see Appendix D for Tables 4-7) report the test accuracy and the number of parameter updates for different datasets and models. First, note the drop in BL accuracy for large batch confirming the accuracy degradation problem. Moreover, note that the FB strategy only works well for moderate batch sizes (it diverges for large batch). However, the GG method has a very consistent performance, but its number of parameter updates are usually greater than our method.

Looking at the last two major columns of Tables 1 and 4-7, the test performances ABS achieves are similar accuracy as BL. Overall, the number of updates of ABS is 3-10 times smaller than BL with batch size 128. However, for most cases, ABSA achieves superior results. This confirms the effectiveness of adversarial training combined with the second order information.

**Table 1:** *Accuracy and the number of parameter updates of C3 on Cifar-10.*

|  | BL | | FB | | GG | | ABS | | ABSA | |
|---|---|---|---|---|---|---|---|---|---|---|
| BS | Acc. | # Iters | Acc. | # Iters | Acc. | # Iters | Acc. | # Iters | Acc. | # Iters |
| 128 | 92.02 | 78125 | N.A. | N.A. | N.A. | N.A. | N.A. | N.A. | N.A. | N.A. |
| 256 | 91.88 | 39062 | 91.75 | 39062 | 91.84 | 50700 | 91.7 | 40792 | **92.11** | 43352 |
| 512 | 91.68 | 19531 | 91.67 | 19531 | 91.19 | 37050 | **92.15** | 32428 | 91.61 | 25388 |
| 1024 | 89.44 | 9766 | 91.23 | 9766 | 91.12 | 31980 | 91.61 | 17046 | **91.66** | 23446 |
| 2048 | 83.17 | 4882 | 90.44 | 4882 | 89.19 | 30030 | 91.57 | 21579 | **91.61** | 14027 |
| 4096 | 73.74 | 2441 | 86.12 | 2441 | 91.83 | 29191 | 91.91 | 18293 | **92.07** | 21909 |
| 8192 | 63.71 | 1220 | 64.91 | 1220 | 91.51 | 28947 | 91.77 | 22802 | **91.81** | 16778 |
| 16384 | 47.84 | 610 | 32.57 | 610 | 90.19 | 28828 | **92.12** | 17485 | 91.97 | 24361 |

## 4.2 ABSA for TinyImageNet and ImageNet

SVHN is a very simple dataset, and Cifar-10/100 are relatively small datasets, and one might wonder whether the improvements we reported in section 4.1 hold for more complex problems. Here, we report the ABSA method on more challenging datasets, i.e., TinyImageNet and ImageNet. We use the exact same hyper-parameters in our algorithm, even though tuning them could potentially be preferable for us.

TinyImageNet is an image classification problem, with 200 classes and only 500 images per class. Thus it is easy to overfit the training data. The results for I1 model is reported in Table 2. Note that with fewer SGD iterations, ABSA can achieve better test accuracy than other methods. The performance of ABSA is actually about 1% higher ( the training loss and test performance of I1 on TinyImagenet is shown in Figure 4 in appendix).

**Table 2:** *Accuracy and the number of parameter updates of I1 on TinyImageNet.*

|  | BL | | FB | | GG | | ABSA | |
|---|---|---|---|---|---|---|---|---|
| BS | Acc. | # Iters | Acc. | # Iters | Acc. | # Iters | Acc. | # Iters |
| 128 | 60.41 | 93750 | N.A. | N.A. | N.A. | N.A. | N.A. | N.A. |
| 256 | 58.24 | 46875 | 59.82 | 46875 | 60.31 | 70290 | **61.28** | 60684 |
| 512 | 57.48 | 23437 | 59.28 | 23437 | 59.94 | 58575 | **60.55** | 51078 |
| 1024 | 54.14 | 11718 | 59.62 | 11718 | 59.72 | 52717 | **60.72** | 19011 |
| 2048 | 50.89 | 5859 | 59.18 | 5859 | 59.82 | 50667 | **60.43** | 17313 |
| 4096 | 40.97 | 2929 | 58.26 | 2929 | 60.09 | 49935 | **61.14** | 22704 |
| 8192 | 25.01 | 1464 | 16.48 | 1464 | 60.00 | 49569 | **60.71** | 22334 |
| 16384 | 10.21 | 732 | 0.5 | 732 | 60.37 | 48995 | **60.71** | 20348 |

ImageNet classification task is perhaps among the most challenging classification problems. Due to the limited computational resources, we only test ABSA and BL, and report results in Figure 5 (see Appendix D). The BL uses $450,045$ parameter updates, reaching $56.32\%$ validation accuracy. For ABSA, the final validation accuracy is $56.40\%$, with only $76,247$ parameter updates. The maximum batch size reached by ABSA is $16,384$, with initial batch size $256$.

Figure 2 shows the result of I3 model on ImageNet. The BL uses $450,045$ parameter updates, reaching $70.46\%$ validation accuracy. The final validation accuracy of ABS and ABSA are $70.15\%$

and $70.24\%$, respectively, both with $66,393$ parameter updates. The maximum batch size reached by ABS and ABSA is $16,384$ with initial batch size $256$. If GG schedule is implemented, the total number of parameter updates would have been $166,216$. (Due to the limitation of resource, we do not run GG for I3 on ImageNet.)

Note that we do not tune the hyper-parameters, e.g., $\alpha, \beta$, and perhaps one could close the gap between $70.24\%$ and $70.4\%$ with fine tuning of our hyper-parameters. However, from a practical point of view such tuning is antithetical to the goal of large batch size training as it would increase the total training time, and we specifically did not want to tailor any new parameters for a particular model/dataset.

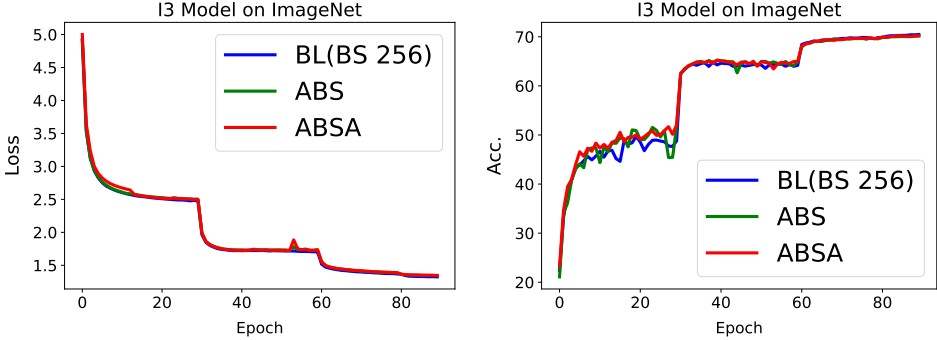

**Figure 2:** *I3 model on ImageNet. Training set loss (left), and testing set accuracy (right), evaluated as a function of epochs.*

### 4.3 APPROXIMATE HESSIAN

One of the limitations of our ABS (ABSA) method is the additional computational cost for computing the top Hessian eigenvalue. If we use the full Hessian operator, the second backpropagation needs to be done all the way to the first layer of NN. For deep networks this could lead to high cost. Here, we empirically explore whether we could use approximate second order information, and in particular we test a block Hessian approximation Figure 6. The block approximation corresponds to only analyzing the Hessian of the last few layers.

In Figure 6 (see Appendix D), we plot the trace of top eigenvalues of full Hessian and block Hessian for C1 model. Although the top eigenvalue of block Hessian has more variance than that of full Hessian, the overall trends are similar for C1. The test performance of C1 on Cifar-10 with block Hessian is $84.82\%$ with 4600 parameter updates (as compared to $84.42\%$ for full Hessian ABSA). The test performance of C4 on Cifar-100 with block Hessian is $68.01\%$ with 12500 parameter updates (as compared to $68.43\%$ for full Hessian ABSA). These results suggest that using a block Hessian to estimate the trend of the full Hessian might be a good choice to overcome computation cost, but a more detailed analysis is needed.

### 5 CONCLUSION

We introduce an adaptive batch size algorithm based on Hessian information to speed up the training process of NNs, and we combine this approach with adversarial training (which is a form of robust optimization, and which could be viewed as a regularization term for large batch training). We extensively test our method on multiple datasets (SVHN, Cifar-10/100, TinyImageNet and ImageNet) with multiple NN models (AlexNet, ResNet, Wide ResNet and SqueezeNext). As the goal of large batch is to reduce training time, we did not perform any hyper-parameter tuning to tailor our method for any of these tests. Our method allows one to increase batch size and learning rate automatically, based on Hessian information. This helps significantly reduce the number of parameter updates, and it achieves superior generalization performance, without the need to tune any of the additional hyper-parameters. Finally, we show that a block Hessian can be used to approximate the trend of the full Hessian to reduce the overhead of using second-order information. These improvements are useful to reduce NN training time in practice.

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

## A    PROOF OF THEOREM

For a finite sum objective function $L(\theta)$, i.e., equation 1, we assume that:

**Assumption 2.** *The objective function $L(\theta)$ satisfies:*

- *$L(\theta)$ is continuously differentiable and the gradient function of $L$ is Lipschitz continuous with Lipschitz constant $L_g$, i.e.*

$$\|\nabla L(\theta_1) - \nabla L(\theta_2)\| \leq L_g\|\theta_1 - \theta_2\|, \qquad \text{for all } \theta_1 \text{ and } \theta_2. \tag{6}$$

- *$L(\theta)$ is strongly convex, i.e., there exists a constant $c_s > 0$ s.t.*

$$L(\theta_2) \geq L(\theta_1) + \nabla L(\theta_1)^T(\theta_2 - \theta_1) + \frac{1}{2}c_s\|\theta_1 - \theta_2\|^2, \qquad \text{for all } \theta_1 \text{ and } \theta_2. \tag{7}$$

  *Also, the global minima of $L(\theta)$ is achieved at $\theta_*$ and $L(\theta_*) = L_*$.*
- *Each gradient of each individual $l_i(z_i)$ is an unbiased estimation of the true gradient, i.e.*

$$\mathbb{E}[\nabla l_i(z_i, \theta)] = \nabla L(\theta), \qquad \text{for all } i. \tag{8}$$

- *There exist scalars $M \geq 0$ and $M_v \geq 0$ s.t.*

$$\mathbb{V}(\nabla l_i(z_i, \theta)) \leq M + M_v\|\nabla L(\theta)\|, \qquad \text{for all } i, \tag{9}$$

  *where $\mathbb{V}(\cdot)$ is the variance operator, i.e.*

$$\mathbb{V}(\nabla l_i(z_i, \theta)) = \mathbb{E}[\|\nabla l_i(z_i, \theta)\|^2] - \|\mathbb{E}[\nabla l_i(z_i, \theta)]\|^2.$$

From the Assumption 2, it is not hard to get,

$$\mathbb{E}[\|\nabla l_i(z_i, \theta)\|^2] \leq M + M_g\|\nabla L(\theta)\|^2, \tag{10}$$

with $M_g = M_v + 1$.

With Assumption 2, the following two lemmas could be found in any optimization reference, e.g. Bottou et al. (2018). We give the proofs here for completeness.

**Lemma 3.** *Under Assumption 2, after one iteration of stochastic gradient update with step size $\eta_t$ at $\theta_t$, we have*

$$\mathbb{E}[L(\theta_{t+1})] - L(\theta_t) \leq -(1 - \frac{1}{2}\eta_t L_g M_g)\eta_t\|\nabla L(\theta_t)\|^2 + \frac{1}{2}\eta_t^2 L_g M, \tag{11}$$

*where $\theta_{t+1} = \theta_t - \eta_t \nabla l_i(\theta, z_i)$ for some $i$.*

*Proof.* With the $L_g$ smooth of $L(\theta)$, we have

$$\mathbb{E}[L(\theta_{t+1})] - L(\theta_t) \leq -\eta_t \nabla L(\theta_t)\mathbb{E}[\nabla l_i(\theta, z_i)] + \frac{1}{2}\eta_t^2 L_g\mathbb{E}[\|\nabla l_i(\theta, z_i)\|^2]$$

$$\leq -\eta_t\|\nabla L(\theta_t)\|^2 + \frac{1}{2}\eta_t^2 L_g(M + M_g\|\nabla L(\theta_t)\|^2).$$

From above, the result follows. □

**Lemma 4.** *Under Assumption 2, for any $\theta$, we have*

$$2c_s(L(\theta) - L_*) \leq \|\nabla L(\theta)\|^2. \tag{12}$$

*Proof.* Let

$$h(\bar\theta) = L(\theta) + \nabla L(\theta)^T(\bar\theta - \theta) + \frac{1}{2}c_s\|\bar\theta - \theta\|^2.$$

Then $h(\bar\theta)$ has a unique global minima at $\bar\theta_* = \theta - \frac{1}{c_s}\nabla L(\theta)$ with $h(\bar\theta_*) = L(\theta) - \frac{1}{2c_s}\|\nabla L(\theta)\|^2$. Using the strong convexity of $L(\theta)$, it follows

$$L(\theta_*) \geq L(\theta) + \nabla L(\theta)^T(\theta_* - \theta) + \frac{1}{2}c_s\|\theta - \theta_*\|_2^2 = h(\bar\theta_*) = L(\theta) - \frac{1}{2c_s}\|\nabla L(\theta)\|^2.$$

□

The following lemma is trivial, we omit the proof here.

**Lemma 5.** *Let $L_B(\theta) = \frac{1}{|B|}\sum_{z_i \in B} l_i(\theta, z_i)$. Then the variance of $\nabla L_B(\theta)$ is bounded by*

$$\mathbb{V}(\nabla L_B(\theta)) \leq M/|B| + M_v\|\nabla L(\theta)\|/|B|, \qquad \text{for all } B. \tag{13}$$

PROOF OF THEOREM 1

Given these lemmas, we now proceed with the proof of Theorem 1.

*Proof.* Assume the batch used at step t is $b_t$, according to Lemma 3 and 5,

$$
\begin{aligned}
\mathbb{E}[L(\theta_{t+1})] - L(\theta_t) &\leq -(1 - \frac{1}{2}b_t\eta_0 L_g(\frac{M_v}{b_t} + 1))b_t\eta_0\|\nabla L(\theta_t)\|^2 + \frac{1}{2}(b_t\eta_0)^2 L_g\frac{M}{b_t} \\
&\leq -(1 - \frac{1}{2}\eta_0 L_g(M_v + b_t))b_t\eta_0\|\nabla L(\theta_t)\|^2 + \frac{1}{2}b_t\eta_0^2 L_g M \\
&\leq -(1 - \frac{1}{2}\eta_0 L_g(M_v + B_{\max}))b_t\eta_0\|\nabla L(\theta_t)\|^2 + \frac{1}{2}b_t\eta_0^2 L_g M \\
&\leq -\frac{1}{2}b_t\eta_0\|\nabla L(\theta_t)\|^2 + \frac{1}{2}b_t\eta_0^2 L_g M \\
&\leq -b_t\eta_0 c_s(L(\theta_t) - L_*) + \frac{1}{2}b_t\eta_0^2 L_g M,
\end{aligned}
$$

where the last inequality is from Lemma 4. This yields

$$
\begin{aligned}
\mathbb{E}[L(\theta_{t+1})] - L_* &\leq L(\theta_t) - b_t\eta_0 c_s(L(\theta_t) - L_*) + \frac{1}{2}b_t\eta_0^2 L_g M - L_* \\
&= (1 - b_t\eta_0 c_s)(L(\theta_t) - L_*) + \frac{1}{2}b_t\eta_0^2 L_g M.
\end{aligned}
$$

It is not hard to see,

$$
\mathbb{E}[L(\theta_{t+1})] - L_* - \frac{\eta_0 L_g M}{2c_s} \leq (1 - b_t\eta_0 c_s)(L(\theta_t) - L_* - \frac{\eta_0 L_g M}{2c_s}),
$$

which concludes

$$
\mathbb{E}[L(\theta_{t+1})] - L_* - \frac{\eta_0 L_g M}{2c_s} \leq \prod_{k=1}^{t}(1 - b_k\eta_0 c_s)(L(\theta_0) - L_* - \frac{\eta_0 L_g M}{2c_s}).
$$

Therefore,

$$
\mathbb{E}[L(\theta_{t+1})] - L_* \leq \prod_{k=1}^{t}(1 - b_k\eta_0 c_s)(L(\theta_0) - L_* - \frac{\eta_0 L_g M}{2c_s}) + \frac{\eta_0 L_g M}{2c_s}.
$$

$\square$

We show a toy example of binary logistic regression on mushroom classification dataset[2]. We split the whole dataset to 6905 for training and 1819 for validation. $\eta_0 = 1.2$ for SGD with batch size 100 and full gradient descent. We set $100 \leq b_t \leq 3200$ for our algorithm, i.e. ABS. Here we mainly focus on the training losses of different optimization algorithms. The results are shown in Figure 3. In order to see if $\eta_0$ is not an optimal step size of full gradient descent, we vary $\eta_0$ for full gradient descent; see results in Figure 3.

## B    OUTLINE OF TRAINING

In this section, we give the detailed outline of our training datasets, models, strategy as well as hyper-parameter used in Alg 1.

**Dataset.** We consider the following datasets.

- **SVHN.** The original SVHN (Netzer et al., 2011) dataset is small. However, in this paper, we choose the additional dataset, which contains more than 500k samples, as our training dataset.

---

[2]https://www.kaggle.com/uciml/mushroom-classification

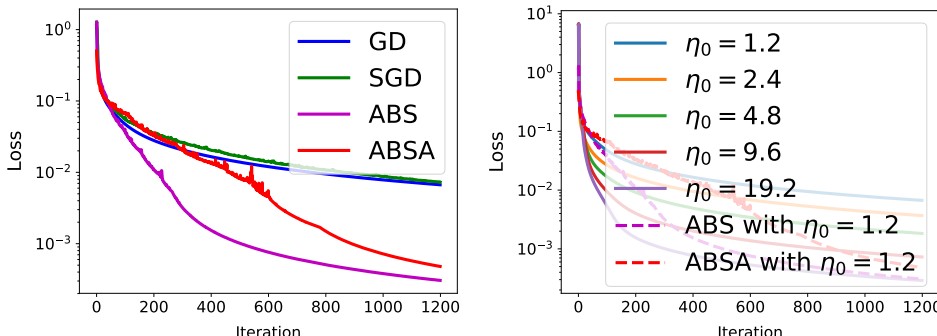

**Figure 3:** *Logistic regression model result. The left figure shows the training loss as a function of iterations for full gradient, SGD, ABS and ABSA. The right figure shows the result of ABS/ABSA compared to full gradient with different learning rate.*

- **Cifar.** The two Cifar (i.e., Cifar-10 and Cifar-100) datasets (Krizhevsky & Hinton, 2009) have same number of images but different number of classes.
- **TinyImageNet.** TinyImageNet consists of a subset of ImangeNet images (Deng et al., 2009), which contains 200 classes. Each of the class has 500 training and 50 validation images.[3] The size of each image is $64 \times 64$.
- **ImageNet.** The ILSVRC 2012 classification dataset (Deng et al., 2009) consists of 1000 images classes, with a total of 1.2 million training images and 50,000 validation images. During training, we crop the image to $224 \times 224$.

**Model Architecture.** We implement the following convolution NNs. When we use data augmentation, it is exactly same the standard data augmentation scheme as in the corresponding model.

- **S1.** AlexNet like model on SVHN as same as Yao et al. (2018)[C1]. We training it for 20 epochs with initial learning rate 0.01, and decay a factor of 5 at epoch 5, 10 and 15. There is no data augmentation.
- **C1.** ResNet18 on Cifar-10 dataset (He et al., 2016). We training it for 90 epochs with initial learning rate 0.1, and decay a factor of 5 at epoch 30, 60, 80. There is no data augmentation.
- **C2.** WResNet 16-4 on Cifar-10 dataset (Zagoruyko & Komodakis, 2016). We training it for 90 epochs with initial learning rate 0.1, and decay a factor of 5 at epoch 30, 60, 80. There is no data augmentation.
- **C3.** SqueezeNext on Cifar-10 dataset (Gholami et al., 2018b). We training it for 200 epochs with initial learning rate 0.1, and decay a factor of 5 at epoch 60, 120, 160. Data augmentation is implemented.
- **C4.** ResNet18 on Cifar-100 dataset (He et al., 2016). We training it for 160 epochs with initial learning rate 0.1, and decay a factor of 10 at epoch 80, 120. Data augmentation is implemented.
- **I1.** ResNet50 on TinyImageNet dataset (He et al., 2016). We training it for 120 epochs with initial learning rate 0.1, and decay a factor of 10 at epoch 60, 90. Data augmentation is implemented.
- **I2.** AlexNet on ImageNet dataset (Krizhevsky et al., 2012). We training it for 90 epochs with initial learning rate 0.01, and decay it to 0.0001 quadratically at epoch 60, then keeps it as 0.0001 for the rest 30 epochs. Data augmentation is implemented.
- **I3** ResNet18 on ImageNet dataset (He et al., 2016). We training it for 90 epochs with initial learning rate 0.1, and decay a factor of 10 at epoch 30, 60 and 80. Data augmentation is implemented.

**Training Strategy.** We use the following training strategies

- **BL.** Use the standard training procedure.
- **FB.** Use linear scaling rule (Goyal et al., 2017) with warm-up stage.
- **GG.** Use increasing batch size instead of decay learning rate (Smith et al., 2017).
- **ABS.** Use our adaptive batch size strategy *without* adversarial training.
- **ABSA.** Use our adaptive batch size strategy *with* adversarial training.

---

[3]In some papers. this validation set is sometimes referred to as a test set.

For adversarial training, the adversarial data are generated using Fast Gradient Sign Method (FGSM) (Goodfellow et al., 2014). The hyper-parameters in Alg. 1 ($\alpha$ and $\beta$) are chosen to be 2, $\kappa = 10$, $\epsilon_{adv} = 0.005$, $\gamma = 20\%$, and $\omega = 2$ for all the experiments. The only change is that for SVHN, the frequency to compute Hessian information is 65536 training examples as compared to one epoch, due to the small number of total training epochs (only 20).

## C  SIMULATED TRAINING TIME

As discussed above, the number of SGD updates does not necessarily correlate with wall-clock time, and this is particularly the case because our method require Hessian backpropagation. Here, we use the method suggested in Gholami et al. (2018a), to approximate the wall-clock time of our algorithm when utilizing $p$ parallel processes. For the ring algorithm Thakur et al. (2005), the communication time per SGD iteration for $p$ processes is:

$$T_{comm} = 2(\alpha_{latency} \log(p) + \beta_{bandwidth} \frac{p-1}{p} |\theta|), \tag{14}$$

where $\alpha_{latency}$ is the network latency, $\beta_{bandwidth}$ is the inverse bandwidth, and $|\theta|$ is the size number of model parameters measured in terms of Bits. Moreover, we manually measure the wall-clock time of computing the Hessian information using our in-house code, as well as the cost of forward/backward calculations on a V100 GPU. The total time will consists of this computation time and the communication one along with Hessian computation overhead (if any). Therefore we have:

$$T_{total} = T_{comp} + T_{comm} + T_{Hess}, \tag{15}$$

where $T_{compute}$ is the time to compute forward and backward propagation, $T_{communication}$ is the time to communicate between different machine, and $T_{Hessian}$ is the time to compute top eigenvalues.

We use the latency and bandwidth values of $\alpha_{latency} = 2 \ \mu s$, and $\beta_{bandwidth} = \frac{1}{6 \ Gb/s}$ based on NERSC's Cori2 supercomputing platform. Based on above formulas, we give an example of simulated computation time cost of I3 on ImageNet. Note that for large processes and small latency terms, the communication time formula is simplified as,

$$T_{comm} = 2\beta_{bandwidth} |\theta|. \tag{16}$$

In Table 3 we report the simulation time of I3 on ImageNet on 512 processes. For GG, we assume it increases batch size by a factor of 10 at epoch 30, 60 and 80. The batch size per GPU core is set to 16 for SGD (and 8 for Hessian computation due to memory limit) and the total batch size used for Hessian computation is set to 4096 images. The $T_{comp}$ and $T_{comm}$ is for one SGD update and $T_{Hessian}$ is for one complete Hessian eigenvalue computation (including communication for Hessian computation). Note that the total Hessian computation time for ABS/ABSA is only $1.15 \times 90 = 103.5 \ s$ even though the Hessian computation is not efficiently implemented in the existing frameworks. Note that even with the additional Hessian overhead ABS/ABSA is still much faster than BL (and these numbers are with an in-house and not highly optimized code for Hessian computations). We furthermore note that we have added the additional computational overhead of adversarial computations to the ABSA method.

**Table 3:** *Below we present the breakdown of one SGD update training time in terms of forward/backwards computation ($T_{comp}$), one step communication time ($T_{comm}$), one total Hessian spectrum computation (if any $T_{Hess}$), and the total training time. The results correspond to I3 model on ImageNet (for accuracy results please see Figure 2).*

| Method | $T_{comp}$ | $T_{comm}$ | $T_{Hess}$ | Total Time |
|--------|-----------|-----------|-----------|------------|
| BL | 2.2E-2 | 1.5E-2 | 0. | 16666 |
| GG | 2.2E-2 | 1.5E-2 | 0. | 6150 (2.71× faster) |
| ABS | 2.2E-2 | 1.5E-2 | 1.15 | 2666 (6.25× faster) |
| ABSA | 3.6E-2 | 1.5E-2 | 1.15 | 3467 (4.80× faster) |

# D ADDITIONAL EMPIRICAL RESULTS

In this section, we present additional empirical results.

**Table 4:** *Accuracy and the number of parameter updates of S1 on SVHN.*

|        | BL    |         | FB    |         | GG    |         | ABS   |         | ABSA    |         |
|--------|-------|---------|-------|---------|-------|---------|-------|---------|---------|---------|
| BS     | Acc.  | # Iters | Acc.  | # Iters | Acc.  | # Iters | Acc.  | # Iters | Acc.    | # Iters |
| 128    | 94.90 | 81986   | N.A.  | N.A.    | N.A.  | N.A.    | N.A.  | N.A.    | N.A.    | N.A.    |
| 512    | 94.76 | 20747   | 95.24 | 20747   | 95.49 | 51862   | 95.65 | 25353   | **95.72** | 24329  |
| 2048   | 95.17 | 5186    | 95.00 | 5186    | 95.59 | 45935   | 95.51 | 10562   | **95.82** | 16578  |
| 8192   | 93.73 | 1296    | 19.58 | 1296    | **95.70** | 44407 | 95.56 | 14400   | 95.61   | 7776    |
| 32768  | 91.03 | 324     | 10.0  | 324     | 95.60 | 42867   | 95.60 | 7996    | **95.90** | 12616  |
| 131072 | 84.75 | 81      | 10.0  | 81      | 95.58 | 42158   | 95.61 | 11927   | **95.92** | 11267  |

**Table 5:** *Accuracy and the number of parameter updates of C1 on Cifar-10.*

|        | BL    |         | FB    |         | GG    |         | ABS   |         | ABSA    |         |
|--------|-------|---------|-------|---------|-------|---------|-------|---------|---------|---------|
| BS     | Acc.  | # Iters | Acc.  | # Iters | Acc.  | # Iters | Acc.  | # Iters | Acc.    | # Iters |
| 128    | 83.05 | 35156   | N.A.  | N.A.    | N.A.  | N.A.    | N.A.  | N.A.    | N.A.    | N.A.    |
| 640    | 81.01 | 7031    | **84.59** | 7031 | 83.99 | 16380   | 83.30 | 10578   | 84.52   | 9631    |
| 3200   | 74.54 | 1406    | 78.70 | 1406    | 84.27 | 14508   | 83.33 | 6375    | **84.42** | 5168   |
| 5120   | 70.64 | 878     | 74.65 | 878     | 83.47 | 14449   | 83.83 | 6575    | **85.01** | 6265   |
| 10240  | 68.75 | 439     | 30.99 | 439     | 83.68 | 14400   | 83.56 | 5709    | **84.29** | 7491   |
| 16000  | 67.88 | 281     | 10.00 | 281     | 84.00 | 14383   | 83.50 | 5739    | **84.24** | 5357   |

**Table 6:** *Accuracy and the number of parameter updates of C2 on Cifar-10.*

|        | BL    |         | FB    |         | GG    |         | ABS   |         | ABSA    |         |
|--------|-------|---------|-------|---------|-------|---------|-------|---------|---------|---------|
| BS     | Acc.  | # Iters | Acc.  | # Iters | Acc.  | # Iters | Acc.  | # Iters | Acc.    | # Iters |
| 128    | 87.64 | 35156   | N.A.  | N.A.    | N.A.  | N.A.    | N.A.  | N.A.    | N.A.    | N.A.    |
| 640    | 86.20 | 7031    | 87.9  | 7031    | 87.84 | 16380   | 87.86 | 10399   | **89.05** | 10245  |
| 3200   | 82.59 | 1406    | 73.2  | 1406    | 87.59 | 14508   | 88.02 | 5869    | **89.04** | 4525   |
| 5120   | 81.40 | 878     | 63.27 | 878     | 87.85 | 14449   | 87.92 | 7479    | **88.64** | 5863   |
| 10240  | 79.85 | 439     | 10.00 | 439     | 87.52 | 14400   | 87.84 | 5619    | **89.03** | 3929   |
| 16000  | 81.06 | 281     | 10.00 | 281     | 88.28 | 14383   | 87.58 | 9321    | **89.19** | 4610   |

**Table 7:** *Accuracy and the number of parameter updates of C4 on Cifar-100.*

|        | BL    |         | FB    |         | GG    |         | ABS   |         | ABSA    |         |
|--------|-------|---------|-------|---------|-------|---------|-------|---------|---------|---------|
| BS     | Acc.  | # Iters | Acc.  | # Iters | Acc.  | # Iters | Acc.  | # Iters | Acc.    | # Iters |
| 128    | 67.67 | 62500   | N.A.  | N.A.    | N.A.  | N.A.    | N.A.  | N.A.    | N.A.    | N.A     |
| 256    | 67.12 | 31250   | **67.89** | 31250 | 66.79 | 46800   | 67.71 | 33504   | 67.32   | 33760   |
| 512    | 66.47 | 15625   | 67.83 | 15625   | 67.74 | 39000   | 67.68 | 32240   | **67.87** | 24688  |
| 1024   | 64.7  | 7812    | 67.72 | 7812    | 67.17 | 35100   | 65.31 | 22712   | **68.03** | 13688  |
| 2048   | 62.91 | 3906    | 67.93 | 3906    | 67.76 | 33735   | 64.69 | 25180   | **68.43** | 12103  |

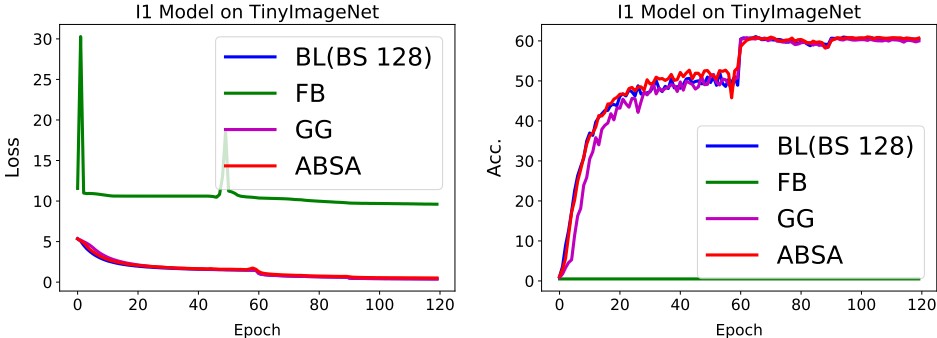

**Figure 4:** *I1 model on TinyImageNet. Training set loss (left), and testing set accuracy (right), evaluated as a function of epochs. All results correspond to batch size 16384 (please see Table 2 for details). As one can see, from epoch 60 to 80, the test performance drops due to overfitting. However, ABSA achieves the best performance with apparently less overfitting (it has higher training loss).*

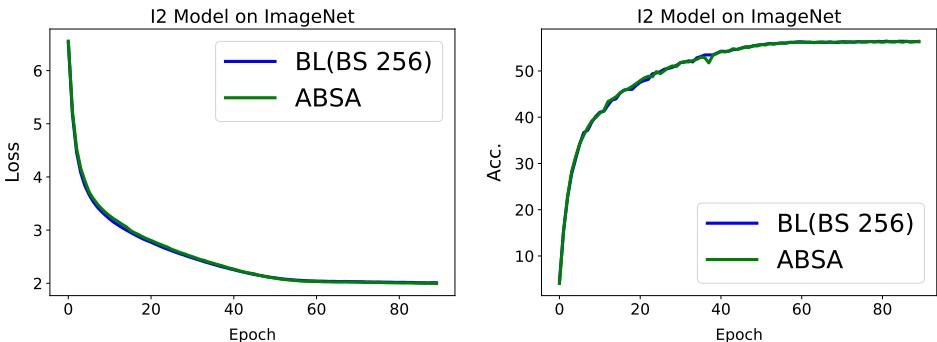

**Figure 5:** *I2 model on ImageNet. Training set loss (a), and testing set accuracy (b), evaluated as a function of epochs.*

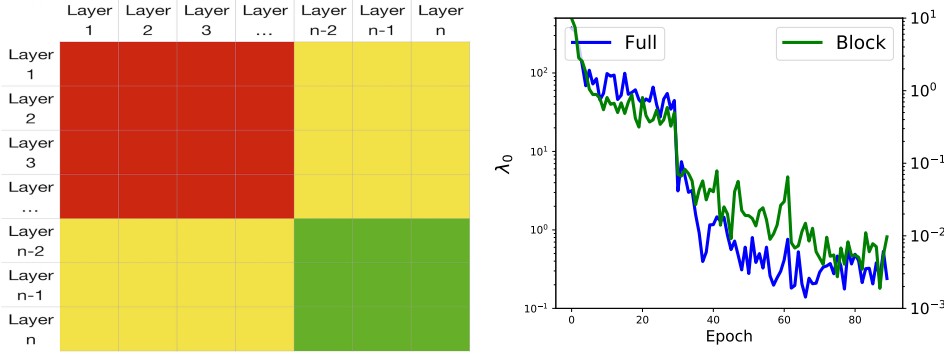

**Figure 6:** *Illustration of block Hessian (left). Instead computing the top eigenvalue of whole Hessian, we just compute the eigenvalue of the green block. .Top eigenvalues of Block of C1 (right) on Cifar-10. The block Hessian is computed by the last two layers of C1. The maximum batch size of C1 is 16000. The full Hessian is based on BL with batch size of 128.*

