# OpenReview forum: "LARGE BATCH SIZE TRAINING OF NEURAL NETWORKS WITH ADVERSARIAL TRAINING AND SECOND-ORDER INFORMATION"
_ICLR.cc/2019/Conference_

### Official Review · AnonReviewer1 · 2018-10-17
**Promising numerical results, but lacks clear description and explanation of the algorithm**

**Rating:** 4
**Confidence:** 4

**Review:**

Based on my understanding, this paper describes a novel approach for addressing the large batch training problem. The authors propose increasing the batch size based on reductions in the largest eigenvalue of the Hessian. This is combined with adversarial training using the fast gradient sign method to reduce the total number of iterations required for training and improve generalization performance. Unfortunately, although the numerical results seem quite promising, the algorithm and its explanation and details are not described clearly in the paper, which makes me lean towards rejection. I describe this more fully below:

1. Description of the Algorithm

The description of the algorithm in Section 3.1 is simply not clear, and lacks clear exposition motivating why the algorithm ought to work. To add to this confusion, there appear to be some inconsistencies between the (brief) description of the method and the description given in the Main Contributions and Limitations section in the Introduction.

As an example, in Section 3.1, the approach for computing the eigenvalue of the Hessian is not described. Which eigenvalue is computed? How is this done? What is the batch size used in this computation? Is it computed over the full training set? The Limitations section briefly describes this (power iteration to tolerance <= 10^-2), but this should be elaborated on in Section 3.1. In fact, the limitations should not be discussed until a clear description of the algorithm is given.

The introduction makes this even more confusing by claiming the second order information is computed by “backpropagating the Hessian operator”. This seems to imply that the 3rd derivative information is computed for second-order information. Later in the Introduction, the authors claim to use Hessian matvecs to perform the power iteration. I believe that the authors mean that the Hessian-vector product is obtained by differentiating the product g’v (a scalar quantity).

In addition, it was not described how the learning rate is changed in the algorithm. Later in the experiments, none of the additional hyperparameters in the procedure are given, such as the duration factor, kappa, the hyperparameters in the adversarial training, and more. This all ought to be included for completeness.

2. Questions about Details of the Algorithm

If it is indeed the case that the authors are using power iteration to compute the largest eigenvalue, why not use Lanczos method as it typically works better for symmetric matrices? In addition, if the intention was to compute the largest eigenvalue of the Hessian, one must be wary that the power iteration/Lanczos method computes the eigenvalue with largest magnitude (the absolute value of lambda), which may mean that it’s possible that the algorithm is utilizing negative curvature information rather than positive curvature information (particularly in the earlier epochs), which may contradict their intuition based on flat minima. This needs to be addressed.

Secondly, there is no explanation as to why increasing the batch size would lead to consistent decrease in the eigenvalues of the Hessian. This is certainly not true for all optimization problems. Even if the flat minima/sharp minima hypothesis is assumed, is it possible for the iterates after increasing the batch size to still tend towards sharper minimizers after being in a flat region? This intuition and explanation needs to be expanded on (and argued for) in order for the algorithm to make any conceptual sense.

Lastly, why is the duration factor needed to increase the batch size if the eigenvalue condition fails? if the duration factor is removed, how does the batch size evolve? Is it necessary? How is the duration factor tuned?

3. Inconsequential Theoretical Results

The authors also prove a theorem bounding the expected optimality gap with adaptive batch sizes. On closer look, this is a simple adaptation of the result by Bottou, Curtis, and Nocedal [2] and does not utilize any of the algorithmic mechanisms described in the paper. Hence, the theoretical result is not novel, does not provide any additional insight on the algorithm, and could be applied to any adaptive/changing batch size SG algorithm. In my opinion, this ought to be removed. (Assumption 2 is also mentioned in the main paper, but is only described in the Appendix.)

4. Additional Considerations

The paper is missing much work done by Nocedal’s group on increasing batch sizes (some of which utilize the L-BFGS approximation to the Hessian); see [1, 3].

Other relevant work by Sagun, Bengio, and others on large batch training, flat minima, and the Hessian in deep learning ought to be included as well; see [4-7].

Lastly, the algorithm demonstrates some significant improvements on the number of iterations. However, efficiency with respect to epochs is not discussed. It may make sense to plot test loss/error against epochs and batch size against iterations for clarity.

Typos/Grammatical Errors:
- Page 2: Should not state “(We refer to this method as ABS)”, easier to include by including (ABS) after Adaptive Batch Size in the beginning of the bullet point.
- Page 6: Section 4: “information” not “informatino”
- Page 6: Section 4: “the” not “teh”
- Page 7: Section 4.1: “confirms” not “confirming”
- Page 7: Section 4.1: no “a” in “a very consistent performance”

Summary:

Overall, although the paper presents some promising numerical results, it lacks a detailed description and explanation of the algorithm to be worthy of publication. It leaves many aspects of the algorithm open to the reader’s interpretation, and I do not believe I could reproduce the results with the information provided. The manuscript needs significant changes to the detail, structure, and writing before it can be considered for publication.

References:
[1] Bollapragada, Raghu, et al. "A progressive batching L-BFGS method for machine learning." arXiv preprint arXiv:1802.05374(2018).
[2] Bottou, Léon, Frank E. Curtis, and Jorge Nocedal. "Optimization methods for large-scale machine learning." SIAM Review 60.2 (2018): 223-311.
[3] Byrd, Richard H., et al. "Sample size selection in optimization methods for machine learning." Mathematical programming134.1 (2012): 127-155.
[4] Chaudhari, Pratik, et al. "Entropy-sgd: Biasing gradient descent into wide valleys." arXiv preprint arXiv:1611.01838(2016).
[5] Jastrzębski, Stanisław, et al. "DNN's Sharpest Directions Along the SGD Trajectory." arXiv preprint arXiv:1807.05031(2018).
[6] Sagun, Levent, et al. "Empirical Analysis of the Hessian of Over-Parametrized Neural Networks." arXiv preprint arXiv:1706.04454 (2017).
[7] Zhu, Zhanxing, et al. "The Regularization Effects of Anisotropic Noise in Stochastic Gradient Descent." arXiv preprint arXiv:1803.00195 (2018).

---

> ### Author Response · Authors · 2018-11-27
> **Authors' response 1**
>
> We would like to thank the reviewer for taking the time to review our paper. Below we provide response to the points raised:
>
>
> >>> The approach for computing the eigenvalue of the Hessian is not described. Which eigenvalue is computed? How is this done? What is the batch size used in this computation? Is it computed over the full training set? The Limitations section briefly describes this (power iteration to tolerance <= 10^-2), but this should be elaborated on in Section 3.1
>
> Response: We should have made this clear in one place in the paper which we will fix. We use power iteration to compute the top eigenvalue of the Hessian, which is mentioned on page 3. The batch size for eigenvalue computation on ImageNet is 4096 (mentioned in Appendix C). In Figure 6, we also point out the batch size is 128 for Cifar-10.  We acknowledge that the discussion on batch size used for eigenvalue computation is not discussed clearly in one place. We will correct this. The important point is that we do not need to compute the eigenvalue on the full training dataset, because as empirically observed in (arxiv: 1810.01021), one can get a very accurate estimate of the eigenvalue by considering sub-sampled Hessian (at least for the problems that we have considered).
>
> >>> The introduction makes this even more confusing by claiming the second order information is computed by “backpropagating the Hessian operator”.
>
> Response: The reviewer is right. We do mean Hessian-vector product is obtained by differentiating the product of $g’v$. We will correct this in the revision.
>
> >>> In addition, it was not described how the learning rate is changed in the algorithm.
>
> Response: The learning rate follows the baseline annealing method for each test considered, except that we change it based on the Hessian information. In particular, when the eigenvalue reduces by a factor of two, we increase the learning rate and batch size by a factor of two (to keep total noise the same according to discussion in arxiv: 1711.04623). We provide the values for the hyper-parameters mentioned (duration factor, kappa, etc) at the end of Appendix B on page 15, which were fixed for all of the experiments performed.
>
> >>> If it is indeed the case that the authors are using power iteration to compute the largest eigenvalue, why not use Lanczos method as it typically works better for symmetric matrices?
>
> Response: It is possible to use Lanczos. However, the reason we do not use Lanczos is that although the Hessian is mathematically symmetric, we are using single precision which results in numerical error.
>
> >>> It is possible that  the algorithm is utilizing negative curvature information rather than positive curvature information (particularly in the earlier epochs), which may contradict their intuition based on flat minima.
>
> Response: We only use the magnitude of the eigenvalue as a measure of curvature of the landscape. It is certainly possible to have a negative eigenvalue, but we only change batch size based on the magnitude of the eigenvalue. Relatively large eigenvalue means the loss landscape is “sharp”, and vice versa.

---

> > ### Author Response · Authors · 2018-11-27
> > **Authors' response 2**
> >
> >
> > >>> Secondly, there is no explanation as to why increasing the batch size would lead to consistent decrease in the eigenvalues of the Hessian.
> >
> > Response: We kindly note that we did not claim increasing the batch size will decrease Hessian eigenvalues. Actually, we use the Hessian eigenvalue to adaptively change the batch size and learning rate. Also, we did not claim that the eigenvalue of the Hessian should necessarily decrease throughout the training procedure.
> >
> > >>> Even if the flat minima/sharp minima hypothesis is assumed, is it possible for the iterates after increasing the batch size to still tend towards sharper minimizers after being in a flat region?
> >
> > Response: Yes this is theoretically possible.
> >
> > >>> Lastly, why is the duration factor needed to increase the batch size if the eigenvalue condition fails? if the duration factor is removed, how does the batch size evolve? Is it necessary?
> >
> > Response: For a simple quadratic problem, the eigenvalue will not change throughout the training. That is why we added the duration factor. It can, of course, be removed.  But if it is removed, then for cases where the training is converging to a landscape that is close to a quadratic minimum, where Hessian spectrum will not change, the batch size will remain constant, which is not desired.
> >
> > >>> How is the duration factor tuned?
> >
> > Response: We did not tune the duration factor (or any other parameters). We used a duration of 10 epochs for all the experiments. One could tune this too and set it to even a smaller number which may further speed up the convergence.
> >
> > >>> The authors also prove a theorem bounding the expected optimality gap with adaptive batch sizes. On closer look, this is a simple adaptation of the result by Bottou, Curtis, and Nocedal [2].
> >
> > Response: We clearly state the limitations of our theoretical work, and never imply that the the novelty of our paper is in its theory. We provide Theorem 1 to show that our ABS method is actually a converging optimization method for the strong convex problem. This is necessary before this method could be applied to the non-convex problems that we have tested. Furthermore, we have actually cited the Bottou, Curtis, and Nocedal paper in our proof (please see Appendix A).
> >
> > >>> The paper is missing much work done by Nocedal’s group on increasing batch sizes (some of which utilize the L-BFGS approximation to the Hessian); see [1, 3]. Other relevant work by Sagun, Bengio, and others on large batch training, flat minima, and the Hessian in deep learning ought to be included as well; see [4-7].
> >
> > Response: This is certainly an oversight and we will add these references to related work in the revised manuscript.
> >
> > >>> Lastly, the algorithm demonstrates some significant improvements on the number of iterations. However, efficiency with respect to epochs is not discussed.
> >
> > Response:  We are not sure what the reviewer refers to by efficiency. If the question about ``efficiency’’ is related to the training time, then we have actually provided this in Table 3. If it is related to how the accuracy changes during training, we have provided the training and testing accuracy in Figure 2-5 which shows that the proposed method achieves better or same accuracy for same epochs.

---

### Official Review · AnonReviewer2 · 2018-11-03
**interesting work, but the theoretical part is not strong enough**

**Rating:** 7
**Confidence:** 4

**Review:**

This paper studies the large batch size training of neural networks, and incorporates adversarial training and second-order information to improve the efficiency and effectiveness of the proposed algorithm. In particular, the authors use second-order information to automatically generate the step size and batch size in each iteration, and apply adversarial training as a regularization method to improve the test performance. Finally, the authors demonstrate their algorithm and compare it with the baseline algorithms on a wide range of datasets. This paper is clearly written and has the following strength:

1.	This paper proposes an adaptive method for SGD training, which proves its convergence for strongly convex optimization.
2.	This paper incorporates the adversarial training and robust optimization into the adaptive SGD training, and shows that this combination significantly improves the test performance.
3.	The authors perform experiments on different datasets, which show that the proposed method enjoys less training time and higher accuracy when using large batch size.

However, this paper also has the following weakness:

1.	The theoretical analysis is somewhat trivial, and the assumption on the objective function is rather strong, which is not consistent with the nonconvex loss functions that are widely applied in training neural networks.
2.	Theorem 1 provides convergence rate of SGD on strongly convex objective functions. However, the authors do not carefully characterize the learning rate to ensure that the loss function achieve \epsilon-accuracy. Moreover, in order to make the last term in (5) be smaller than \epsilon, the learning rate \eta_0 should be in the order of O(\epsilon), which is no longer a tuning-free parameter.
3.	The authors mention that the proposed algorithm converges faster than basic SGD, but it is not clearly demonstrated from Theorem 1.
4.	I am confused about how to determine the number of iterations for different algorithm as shown in Table 1s and 2? Do you stop each algorithm when they attain the same training error on the training dataset?
5.	It is also confused that the number of iterations for ABS and ABSA are relatively larger than that of BL (Tables 1 and 2), but the training time of BL is longer than those of ABS and ABSA as reported in Table 3?
6.	Some minor flaws. In (9) it should be \|\nabla L(\Theta)\|^2; in Lemma 3, the expectation on the left side should be taken conditioned on \theta_t.

---

> ### Author Response · Authors · 2018-11-27
> **Authors' response**
>
> We would like to thank the reviewer for taking the time to review our paper. Below we provide response to the points raised:
>
>
> >>> The authors do not carefully characterize the learning rate to ensure that the loss function achieve \epsilon-accuracy. Moreover, in order to make the last term in (5) be smaller than \epsilon, the learning rate \eta_0 should be in the order of O(\epsilon), which is no longer a tuning-free parameter.
>
> Response: We kindly note that we have double checked the proof and it is correct. In Theorem 1, we fixed the learning rate for the whole training procedure to show how the error changes as a function of iterations, \textit{not necessarily for an O(\epsilon) error}. If you see the last line of our proof (p. 13 before QED), we have the inequality for how far we are in expectation from L^\star. You can see that to get \epsilon accuracy the learning rate has to decay. We will clarify this in the text.
> Another confusion we would like to clarify concerns the nature of our method. We never claim our method is hyper-parameter free. There is no such optimization method found so far. What we state, a number of times in the paper, is that the hyper-parameter setting of our method is exactly the same for all the diverse experiments that we performed. That is, we did not tailor our hyper-parameters for particular tests to boost the performance of our method, even though our tests involved eight state-of-the-art neural network models on four standard datasets, including ImageNet. We were particularly careful about this, because a solution for large batch size training should not add to the list of hyper-parameters that need to be tuned, as this would actually increase the training time which is antithetical to the goal of large batch size training.
>
> >>> The authors mention that the proposed algorithm converges faster than basic SGD, but it is not clearly demonstrated from Theorem 1.
>
> Response: We discussed this on page 6 right after Theorem 1. This follows from the fact that (1-b_k\eta_0c_s) <  (1-\eta_0c_s). We also show a numerical example in Appendix A to illustrate the faster convergence (see Fig. 3).
>
> >>> Do you stop each algorithm when they attain the same training error on the training dataset?
>
> Response: No, the number of training epochs is fixed, based on the corresponding paper that introduced each neural network. Other criteria such as stopping based on the same training loss or validation could be used as well, but we want to follow exactly the baseline training procedure for each case. Other works on large batch size training have also followed this strategy (please see arxiv: 1711.00489, 1706.02677 and 1708.03888).  Please see Appendix B where we show the detailed training outline for each of the experiments performed.
>
> >>> I am confused about how to determine the number of iterations for different algorithm as shown in Table 1-2?
>
> Response: Table 1 and 2 report the number of SGD iterations required to finish these fixed number of epochs. Also, we plot how the training loss changes for different methods during training in Figure 2, 4, 5.
>
> >>> It is also confused that the number of iterations for ABS and ABSA are relatively larger than that of BL (Tables 1 and 2), but the training time of BL is longer than those of ABS and ABSA as reported in Table 3?
>
> Response: This is a very important confusion, and we will make sure to better explain this in the revision. We need to compare apples to apples, that is, we must compare methods that achieve the same accuracy. Therefore, we need to compare the baseline and ABSA for cases where both achieve the same accuracy. Baseline accuracy significantly drops for large batches, whereas ABSA accuracy does not. So for comparison, we should consider baseline with a batch size that achieves similar accuracy as ABSA, and that only occurs for small batch sizes.
> For instance, the baseline in Table 2 with a batch size of 16K requires 732 SGD iterations but it only achieves 10.21% accuracy, which cannot be compared with our method which achieves >60% accuracy. To properly compare the timings we need to consider baseline with 128 batch size which requires 93K SGD iterations and achieves 60.41% accuracy. This is one of the major points of the paper, and we will make this more clear in the revision.
>
> >>> Some minor flaws. In (9) it should be \|\nabla L(\Theta)\|^2; in Lemma 3, the expectation on the left side should be taken conditioned on \theta _t.
>
> Response: That is correct. We will fix those in the next version.

---

> > ### Comment · AnonReviewer2 · 2018-12-05
> > **Thank you for addressing my comments**
> >
> > The authors have fully addressed my questions. Please update your paper accordingly. Nice work!

---

> > > ### Author Response · Authors · 2018-12-08
> > > **Thanks a lot**
> > >
> > > We would like to thank the reviewer for the detailed review and feedback. We will update the paper accordingly.

---

### Official Review · AnonReviewer3 · 2018-11-07
**Clearly an unfinished paper**

**Rating:** 4
**Confidence:** 5

**Review:**

The authors propose using information from the Hessian to grow the batch size as the training progresses. It is well-known that larger batch sizes can be used for later stages of optimization (ie, https://arxiv.org/abs/1711.00489, https://arxiv.org/abs/1706.05699), but they are missing motivation as to why use Hessian information for this.

Furthermore, the description of the algorithm is lacking detail and is essentially unreproducible in current form.

The main description of their method is Algorithm 1 box, which suggests to grow batch size when "eigenvalue" is much smaller than previous eigenvalue. Is that the top eigenvalue? How is it estimated? Why is that the criterion? Note that for stochastic least squares problem one benefits from later batch sizes in later stages of optimization even though Hessian doesn't change.

In section Section 4.3 they start talking a bit about computing Hessian, referring to non-existent figure 6 for details of block approximation.

Authors mention that Hessian computation is not supported in major frameworks but don't provide explanation of how they compute it (did they not use a major framework for ImageNet experiments?).

Note that a single row of Hessian (hence full Hessian) can be computed in all major frameworks by differentiating an element of the gradient. IE, in PyTorch https://gist.github.com/apaszke/226abdf867c4e9d6698bd198f3b45fb7, and also eigenspectrum of Hessian can be approximated -- https://github.com/noahgolmant/pytorch-hessian-eigenthings

---

> ### Author Response · Authors · 2018-11-27
> **Authors' response**
>
> We would like to thank the reviewer for taking the time to review our paper. Below we provide response to the points raised:
>
>
> >>> The main description of their method is Algorithm 1 box, which suggests to grow batch size when "eigenvalue" is much smaller than the previous eigenvalue. Is that the top eigenvalue? How is it estimated?
>
> Response: Yes, that is correct, we compute the top Hessian eigenvalue using power iteration. We will clarify this in the revised manuscript. We mentioned in Algorithm 1, when the eigenvalue decays by a factor of 2, the batch size increases proportionally.
>
> >>> Why is that [Hessian spectrum] the criterion?
>
> Response: The main intuition is to increase batch size only in areas where the loss landscape is flatter. That is we have empirically found that using a larger batch size in regions where the loss has a “flatter” landscape, and using a smaller batch size in regions with a “sharper” loss landscape, is very effective in avoiding attraction to local minima with poor generalization. To be precise by effective, the SGD iterations can reduce up to 5x as compared to existing state-of-the-art for large batch training.
>
> >>> Note that for stochastic least squares problem one benefits from later batch sizes in later stages of optimization even though Hessian doesn't change.
>
> Response: The problem of least square is a good point. That is why we introduce the “Duration Time” in our algorithm to overcome this problem, i.e., we adaptively increase the batch size as the Hessian eigenvalue decreases or stays stable for several epochs (fixed to be ten in all of the experiments). However, we are not claiming that our method increases the batch size in the most possible efficient way. Even without the duration factor, the batch size would remain constant. However, note that our method requires significantly less as compared with other state-of-the-art.
>
> >>> In section Section 4.3 they start talking a bit about computing Hessian, referring to non-existent figure 6 for details of block approximation.
>
> Response: That is not correct. Perhaps the reviewer has missed the appendix. When Figure 6 is used as a reference in section 4.3, we mentioned in the parenthesis that Figure 6 is in Appendix D.
>
> >>> Authors mention that Hessian computation is not supported in major frameworks but don't provide explanation of how they compute it (did they not use a major framework for ImageNet experiments?).
>
> Response: In Limitation, we mentioned that most of the existing frameworks do not support (memory) efficient backpropagation of the Hessian, not that they do not support it. We have developed a pure python library as well as a PyTorch library and we will add a reference to it after double-blind review is finished.
>
> >>> Note that a single row of Hessian (hence full Hessian) can be computed in all major frameworks by differentiating an element of the gradient. IE, in PyTorch https://gist.github.com/apaszke/226abdf867c4e9d6698bd198f3b45fb7, and also eigenspectrum of Hessian can be approximated -- https://github.com/noahgolmant/pytorch-hessian-eigenthings
>
> Response: Unfortunately we cannot completely address this question due to double-blind policy. However, to briefly respond to the question, the first link (https://gist.github.com/apaszke/226abdf867c4e9d6698bd198f3b45fb7) computes the Hessian matrix explicitly. For small-scale problem, one can do that. However, for NNs, there are millions of dimensions, and such approach would be infeasible. We will further address this question after the double-blind period is finished.

---

### Author Response · Authors · 2018-11-27
**Summary Response**

We would like to thank all the reviewers for taking the time to review our work and provide their feedback. Below we provide a general response and then a more detailed response to every reviewer.


>>> The assumption of the theoretical results are too strong

Response: There is confusion regarding the purpose of the theoretical portion of the paper. Our goal in Theorem 1 is merely to show that the ABS method is a converging optimization method for a strongly convex problem. Furthermore, we explained in detail in the limitations section of the paper that this is not the focus of our work and that the assumptions are limited. We will further clarify this in the revised manuscript.

>>> It is already clear that increasing batch size during training can avoid “generation gap” in previous work.

Response: The seminal works of arxiv: 1711.00489, 1712.02029, and 1711.0462 illustrated that batch size could be used to anneal the noise during training. However, two new contributions of our work are: (i) the annealing scheduling proposed in prior work is not necessarily an optimal strategy to achieve fewer SGD iterations, and (ii) we have shown that combining the adaptive batch size with robust optimization results in better generalization, as compared to other large batch methods.

Our method does not require extra hyper-parameter tuning, and we have shown this by the most extensive testing so far on multiple datasets (Cifar-10/100, SVHN, TinyImagenet, ImageNet) and multiple NNs (ResNet, Wide ResNet, SqueezeNet etc).  The results in Tables (1-2,4-8) clearly show that the proposed robust Hessian based method achieves a smaller number of SGD updates (up to 5x) as compared to existing state-of-the-art. We emphasized that this is achieved without any hyper-parameter tuning.  A fair evaluation of this work should be viewed in comparison with the current state-of-the-art.

---

### Meta-Review · Area_Chair1 · 2018-12-13
**not enough support from reviewers to accept this paper**

**Confidence:** 5
**Recommendation:** Reject

**Metareview:**

I would like to commend the authors on their work engaging with the reviewers and for working to improve training time. However, there is not enough support among the reviewers to accept this submission. The reviewers raised several important points about the paper, but I believe there are a few other issues not adequately highlighted in the reviews that prevent this work from being accepted:

1. [premises] It has not been adequately established that "large batch training often times leads to degradation in accuracy" inherently which is an important premise of this work. Reports from the literature can largely be explained by other things in the experimental protocol. Even the framing of this issue has become confused since, although it may be possible to achieve the same accuracy at any batch size with careful tuning, this might require using (at worst) the same number of steps as the smaller batch size in some cases and thus result in little to no speedup. For example see https://arxiv.org/abs/1705.08741 and recent work in https://arxiv.org/abs/1811.03600 for more information. Even Keskar et al. reported that data augmentation eliminated the solution quality difference between their larger batch size and their smaller batch size experiments which indicates that even if noisiness from small batches serving to regularize training other regularization techniques can serve just as well.

2. [baseline strength] The appropriate baseline is standard minibatch SGD w/momentum (or ADAM or whatever) algorithm with extremely careful tuning of *all* of the hyperparameters. None of the popular learning rate heuristics will always work and other optimization parameters need to be tuned as well. If learning rate decay is used, it should also be tuned especially if one is trying to measure a speedup. The submission does not provide a sufficiently convincing baseline.

3. [measurement protocol] The protocol for measuring a speedup is not convincing without more information on how the baselines were tuned to achieve the same accuracy in the fewest steps. Approximating the protocols in https://arxiv.org/abs/1811.03600 would be one alternative.

Additionally there are a variety of framing of issues around hyperparameter tuning, but, because they are easier to fix, they are not as salient for the decision.

---

> ### Author Response · Authors · 2019-01-28
> **Response**
>
>
> >>> I believe there are a few other issues not adequately highlighted in the reviews that prevent this work from being accepted:
>
> Response: We would like to thank the AC for taking the time to post the meta-review.  Unfortunately, the major concerns are simply not correct.  Below we address the points raised:
>
> >>> [premises] It has not been adequately established that "large batch training often times leads to degradation in accuracy" inherently which is an important premise of this work.
>
> Response: That is incorrect. Multiple (if not tens) of works have shown that large batch size leads to significant degradation of accuracy (arxiv: 1609.04836, 1802.08241, 1706.02677,1811.12941, 1811.03600, etc). This is a well established phenomena and it is surprising that the area chair brings this point. Even the papers that the AC cites do clearly mention this.
>
> >>>Even the framing of this issue has become confused since, although it may be possible to achieve the same accuracy at any batch size with careful tuning, this might require using (at worst) the same number of steps as the smaller batch size in some cases and thus result in little to no speedup. For example see https://arxiv.org/abs/1705.08741 and recent work in https://arxiv.org/abs/1811.03600 for more information.
>
> Response: This is incorrect. Large batch does not “at worst” need the same number of steps as smaller batch to achieve the same accuracy. It is not clear to us why the AC would even claim this. Large batch training may not be able to recover baseline if a vanilla SGD is used even with longer training (arxiv:1811.03600 Figure 23).
>
> >>> Even Keskar et al. reported that data augmentation eliminated the solution quality difference between their larger batch size and their smaller batch size experiments which indicates that even if noisiness from small batches serving to regularize training other regularization techniques can serve just as well.
>
> Response: This is completely incorrect and a misunderstanding of Keskar’s paper by the AC. Keskar’s paper (arxiv: 1609.04836) showed that data augmentation can partially alleviate the generation gap between small and large batch training, and it is not possible to completely close this gap with data-augmentation.
>
> >>> [baseline strength] The appropriate baseline is standard minibatch SGD w/momentum (or ADAM or whatever) algorithm with extremely careful tuning of *all* of the hyperparameters. None of the popular learning rate heuristics will always work and other optimization parameters need to be tuned as well. If learning rate decay is used, it should also be tuned especially if one is trying to measure a speedup. The submission does not provide a sufficiently convincing baseline.
>
> Response: This is not applicable to our work. We do NOT do any hyper-parameter tuning, and we emphasized this in multiple places in the paper. This was one of the main points of the paper, and it is very disappointing that it has been missed. Moreover, we did not change any hyper-parameters for the baseline neural networks that we used.
>
> >>> [measurement protocol] The protocol for measuring a speedup is not convincing without more information on how the baselines were tuned to achieve the same accuracy in the fewest steps. Approximating the protocols in https://arxiv.org/abs/1811.03600 would be one alternative.
>
> Response: Please see above, we did not tune any hyper-parameters.
>
>
>
> Also we should mention that Reviewer 1 is pointing to our own code in his review as a reason that our Hessian backpropogation is not novel.  That is, due to the double blind aspect of this publication venue, a reviewer incorrectly thought that our novel Hessian backpropogation procedure was due to someone else and thus not novel.   It was very frustrating for us that we could not clear this due to double blind review policy. Moreover, the reviewer completely missed the supplementary material (thus mentioning non-existent figures). We had hoped that he would at least read our response but unfortunately that did not happen. Combined with AC’s meta review it seems our paper’s points and efforts to clear up confusions were not helpful.
>
> At the end, we would like to specially thank Reviewer 2 and Reviewer 3 for taking the time to follow up with our response, and providing their valuable feedback.